# DVGfinder: A Metasearch Tool for Identifying Defective Viral Genomes in RNA-Seq Data

**DOI:** 10.3390/v14051114

**Published:** 2022-05-23

**Authors:** Maria J. Olmo-Uceda, Juan C. Muñoz-Sánchez, Wilberth Lasso-Giraldo, Vicente Arnau, Wladimiro Díaz-Villanueva, Santiago F. Elena

**Affiliations:** 1Instituto de Biología Integrativa de Sistemas (I^2^SysBio), CSIC-Universitat de València, 46980 Valencia, Spain; mariajose.olmo@csic.es (M.J.O.-U.); jc.munoz@csic.es (J.C.M.-S.); wilasgi@alumni.uv.es (W.L.-G.); vicente.arnau@uv.es (V.A.); wladimiro.diaz@uv.es (W.D.-V.); 2Santa Fe Institute, Santa Fe, NM 87501, USA

**Keywords:** benchmarking, bioinformatics, defective viral genomes, gradient boosting, machine learning, RNA-seq, SARS-CoV-2, virus replication

## Abstract

The generation of different types of defective viral genomes (DVG) is an unavoidable consequence of the error-prone replication of RNA viruses. In recent years, a particular class of DVGs, those containing long deletions or genome rearrangements, has gain interest due to their potential therapeutic and biotechnological applications. Identifying such DVGs in high-throughput sequencing (HTS) data has become an interesting computational problem. Several algorithms have been proposed to accomplish this goal, though all incur false positives, a problem of practical interest if such DVGs have to be synthetized and tested in the laboratory. We present a metasearch tool, DVGfinder, that wraps the two most commonly used DVG search algorithms in a single workflow for the identification of the DVGs in HTS data. DVGfinder processes the results of ViReMa-a and DI-tector and uses a gradient boosting classifier machine learning algorithm to reduce the number of false-positive events. The program also generates output files in user-friendly HTML format, which can help users to explore the DVGs identified in the sample. We evaluated the performance of DVGfinder compared to the two search algorithms used separately and found that it slightly improves sensitivities for low-coverage synthetic HTS data and DI-tector precision for high-coverage samples. The metasearch program also showed higher sensitivity on a real sample for which a set of copy-backs were previously validated.

## 1. Introduction

One of the hallmarks of RNA virus replication is their low fidelity, traditionally attributed to the lack of proofreading activities in their RNA-dependent RNA polymerases (RdRp), although a suite of RNA-modifying activities associated to increased fidelity has been identified in coronaviruses [1]. During viral replication, a plethora of defective copies of the viral genome, or DVGs, can be generated: point mutations and hypermutations, frame-shiftings, short and large deletions, and largely rearranged genomes (e.g., copy- and snap-backs), all forming part of the mutant swarm that constitute the viral population. These RNA molecules are called defective genomes because they need the presence of the wild-type viral genome to complete their replication cycle [2]. Since defective viral particles were first described in influenza A virus by Von Magnus and Gard [3], examples of DVGs have been pervasively reported in positive and negative single-stranded RNA viruses [4] and double-stranded RNA viruses and retroviruses [5]. A particular interesting type of DVG is known as defective interfering particles (DIPs), which show the capacity to encapsulate and interfere with the replication and population dynamics of the wild-type virus [6]. In a recent review, DVGs have been considered as long noncoding RNAs, though they might preserve some coding capacity [7].

Recent studies suggest that DVG generation is not a completely stochastic process and may be mediated by viral and host factors [4]. This new observation opens the possibility of manipulating the generation of DVGs (and DIPs) with therapeutic purposes, resulting in the so-called therapeutic interfering particles (TIPs) [5]. Indeed, the first TIP rationally designed and synthetically produced to interfere with SARS-CoV-2 has already been published [8].

The first virus–host interaction model with production of DIPs was proposed by Huang and Baltimore [9]. More recent models take into account the immune stimulation effect that DIPs could exhibit [4,6]. As mentioned above, DIPs could be used therapeutically as antivirals [10] and vaccine adjuvants [11], and although the specific mechanisms of generation are not well understood yet, their impact in viral evolutionary dynamics has been well stablished, both experimentally and theoretically [5]. Additional information is clearly needed to better appreciate the implications of DVGs in pathogenesis. For instance, to understand the mechanisms behind their apparently antagonistic functions: on the one hand, in most infections DVGs are triggering the host immune system but, on the other hand, evidence exists that involucrate them into viral persistence [12].

The types of DVG frequently considered have been deletions and copy- and snap-backs (Figure 1) but insertions and rearrangements with the host’s mRNAs should not be discarded. Although the terminology is not yet completely unified, the DVGs can be characterized by three parameters: the genome position at which the RdRp is released from the template or *break-point* (BP); the genome position at which the RdRp reattaches to the template and continues polymerizing; or the *reinitation site* (RI) and the sense of the fragments pre- and post-BP/RI junction (Figure 1).

The strategy most often used to identify DVGs in high-throughput sequencing (HTS) data consists of, as a first step, aligning all reads against the reference viral genome and subsequent identification of the BP and RI coordinates. In recent years, some programs have been published pursuing this objective. As far as we know, the earlier program that took into account the DVGs was Paparazzi [13] although it was not specifically designed to identify DVGs and excluded them to reconstruct the consensus viral genome in a sample. The first specific program for DVG identification and characterization in HTS data was ViReMa-a [14], which together with DI-tector [15] are the most widely used algorithms to date. Table 1 lists the different published bioinformatic tools for DVG detection.

Although a rigorous assessment of the comparative performance of these programs has not been done, it is well known that most of these algorithms introduce in silico artifacts (e.g., false positives and false negatives) during its identification and assembly process. For instance, the introduction of in silico artifacts by DI-tector has been previously reported and discussed by Bosman et al. [28]. Likewise, the low sensitivity of ViReMa-a identifying copy-backs has been already mentioned by Beauclair et al. [15].

Here, we present a new machine learning (ML)-based tool, DVGfinder, that facilitates the identification and extraction of a set of candidate DVGs for further experimental validation. For that, our tool integrates the two most commonly used DVG searching algorithms (Table 1), ViReMa-a and DI-tector, in a single workflow unifying the terminology and adding a set of descriptive variables. Additionally, DVGfinder applies an ML algorithm to reduce the number of false positives and generates a user-friendly and rich HTML report with graphical outputs that help a first exploration of the DVG candidates. Although the search results are processed by DVGfinder, the individual output of both search algorithms is also stored, so that users can trace the events predicted by each program’s own terminology.

## 2. Materials and Methods

### 2.1. Implementation, Requirements, and Availability

DVGfinder is written in Python3 and bash and, for now, it runs in the Linux-type command line. It calls the programs ViReMa-a (v.0.23) and DI-tector (v.0.6), the aligners bowtie and bwa, and some specific Python libraries. It is highly recommended to create an environment; for that reason, a configuration environment file, Conda-like formatted (dvgfinder_env.yaml), with all the external programs and Python library requirements is provide in the DVGfinder repository [34].

The parameters needed for execution are -*fq*, sample in fastq format; -*r*, virus reference genome in fasta format, and the indexed genome for bwa and bowtie in the same directory; -*m*, number of bases to take into account in the measurement of the mean depth over the junction BP-RI (default 5); -*t*, probability threshold to filter out the true DVGs from false positives (default 0.5); -*n*, number of processes to be used; and -*s*, polarity of the reference virus (default is “+”).

DVGfinder process single-end short reads (Illumina-like); also, it is recommended to put the reference and their indexed files by bowtie and bwa in the “/ExternalNeeds/reference” directory.

### 2.2. Algorithm

The program is structured in three modules acting sequentially (Figure 2): (1) Metasearch module—launches the search of DVGs in the HTS data with the ViReMa-a and DI-tector algorithms. The program then extracts the structural DVGs identified by the search algorithms and characterizes them by generating informative variables with a unified terminology (see below). (2) Prediction module—applies the trained classification model to the set of identified DVGs and assigns each DVG a probability of being a true positive. (3) Report module—writes final tables, prepares the graphic visualizations, and wraps up everything into an HTML final report.

There are four cases of usage (Figure 2A), and depending on whether the Metasearch module finds the candidate DVGs with both search algorithms, the Prediction module is launched (Figure 2A, Case 1).

### 2.3. Descriptive Features

The last part of the Metasearch module consists of the addition of descriptive variables to each DVG. These features add information about the depth in both BP and RI coordinates, and about the mean depth in its neighborhoods, both in absolute and normalized values, as explained below. The estimated length of the DVGs is also provided. All this information is saved in a csv-formatted text file ({sample_name}_resume_table.csv). The predictor variables, i.e., the specific features used by the predictive model, are metrics related to the proportion of the DVGs in the coordinates BP and RI and in its neighborhood as follows:

1.*RPHT*, or reads per hundred thousand, relativizes the number of reads where the BP/RI junction has been found to the total number of reads that are correctly aligned with the reference sequence (mapped reads, first selection (Figure 2C, 1.2)). The use of *RPHT* was previously documented by Parker et al. [35] and, instead of hundred thousand, relative to million reads mapped (*RPM*) By Razelj et al. [2]:(1)$$RPHT=10^{5}\times DVG\;read\;counts/total\;mapped\;reads$$2.*p*BP and *p*RI, or the proportion of reads in the BP and RI genome coordinates, normalizes the number of reads presenting the DVG with respect to the total number of mapped reads at these positions. The values can be greater than one because the search algorithms can find DVGs in reads that do not align with the reference in the first moment.
(2)pX=DVG read counts/total depth in coordinate X
where *X* ∈ [BP, RI].3.*SDRM* stands for the semi-difference relative to the mean for positions *a* = depth in BP and *b* = depth in RI. *SDRM* can also be computed as the depth mean in the neighborhood of the coordinates of interests *X* and, in this case, *a* = mean pre-coordinate *X* depth and *b* = mean post-coordinate *X* depth:(3)SDRM=|a−b|/(a+b)

These metrics were calculated using the information of the two alignments produced by bowtie and bwa mem, respectively. An additional level of resolution also was considered, separating the hard-clipped reads from the rest of the mapped reads. Henceforth, the Metasearch module incorporates information about *pX* and *SDRM* from the four alignments; i.e., the mapped and hard-clipped from the two alignment algorithms (Figure 2C).

The predictors depend on the output of both search algorithms; thus, the predictive module is only applied in Case 1 in Figure 2A. In Case 1, a consensus level is generated with the DVGs detected by the two search programs.

The final outputs also report descriptive variables such as the sense, BP, and RI coordinates, the type of DVG in which it has been classified, and the theoretical length of the deletion. In addition, we add a unique identifier of the DVG generated with the format “sense_BP_RI”. For example, identifier “++_100_300” represents a deletion in the coding sense involving the fragment from genomic positions 101 to 299 (Figure 1). Likewise, identifier “+−_100_80” would represent a copy-back in which nucleotides between positions 81 to 99 form the loop (Figure 1).

### 2.4. Estimation of the Theoretical Length of DVG

The program calculates the theoretical length of the DVGs’ reconstruction. To do this, two reasonable assumptions have been made. Firstly, it has been considered that a DVG only results from an abnormal junction event. Secondly, once the RdRp reattaches in the RI point, it continues the polymerization of the DVG until running off at the end of its template (Table 2).

### 2.5. Model Generation

A Gradient Boosting Classifier (GBC) was chosen to build the predictive model using the methods implemented by the scikit-learn library [36] and following the DOME recommendations for supervised ML validation in biology [37]. Before selecting this algorithm, we tested several combinations of algorithms and number of predictors and choose the current one based on its best scores in accuracy, precision, *F*_1_ score, receiver operating characteristic curve (*ROC*), and the area under the *ROC* curve (*AURC*). Figure 3 summarizes the process.

#### 2.5.1. Data

The training dataset consists of 1526 events with the categories real and artifact balanced (764 and 762, respectively). This dataset was generated as follows: we created a synthetic fastq file simulating the sequencing of a known set of DVGs (630 different genomes) and its wild-type reference genome, using as reference the (+)ssRNA of SARS-CoV-2 (NC_045512.2). The Metasearch module of DVGfinder was applied on this sample resulting in a dataset with 1526 candidates that were then labeled as real or artifacts; defining as reals the DVGs directly introduced in the set of known DVG and its reverse complement events. The predictors used had been explained in Section 2.3 above. The labeled dataset is accessible in the DVGfinder’s GitHub repository [38].

#### 2.5.2. Training, Optimization, and Evaluation of the Model

The whole synthetic dataset was split into train (75%) and test (25%) sets using the parameter “stratify = y” from “train_test_split”.

Eighteen predictors were selected to train the model and through the optimization of the hyperparameters with the RandomizedSearchCV function, these were adjusted to max_depth = 2, max_features = sqrt, min_samples_leaf = 2, min_samples_split = 4, and n_estimators = 50.

The accuracy, recall, precision, *F*_1_ score, and *AURC* of the model were 100-repeated and 4-fold cross-validated in the training set (Figure 4A) and the optimized model was finally evaluated in the test set (Figure 4B). The values obtained in the learning and the evaluation process were very close, therefore considering that the model was generalizable. The final model was built with the totality of the data (Figure 3).

### 2.6. Performance Metrics

The number of true positives (*TP*), false positives (*FP*), false negatives (*FN*), and true negatives (*TN*) were used to compute the following evaluation metrics:
false positive rate,
(4)FPR=FP/(FP+TN)
precision or positive prediction value,
(5)PPV=TP/(FP+TP)=1−FDR
sensitivity, recall, hit rate, or true positive rate,
(6)TPR=TP/(FN+TP)
and the harmonic mean of PPV and TPR, or
(7)$$F_{1}=2\times PPV\times TPR/\left(PPV+TPR\right)$$
where *FDR* in Equation (5) is the false discovery rate.

### 2.7. Output of DVGfinder

The identification of the DVGs is strand sensitive, so it can be used with this type of sequencing without losing information. In case the sequencing was without preserving the polarity, the reverse complement DVGs would be grouped together, considering for example the deletion forward “++_30_200” the same as the deletion reverse “−−_200_30”.

The results of DVGfinder are written in different files and an interactive HTML report is generated.

#### 2.7.1. HTML Interactive Report

In Case 1 in Figure 2A, the HTML report shows the detected DVGs in up to three levels: (1) All run mode, with all the DVGs detected by the two search algorithms; (2) Consensus run mode, reporting only the DVGs found by both algorithms (sense sensitive); and (3) Filtered run mode, with the DVGs predicted as reals by the ML classifier, with a probability greater than the threshold parameter defined by the user (default is 0.5). Furthermore, the three types of graphics generated—scatter plot of the BP-RI coordinates, distribution of the DVG length, and arc diagram of the deletions—were labeled by the program that had identified them.

In Cases 2 and 3 in Figure 2A, the report consists of a one-page HTML with the same data structures as in Case 1 but only with the results of the algorithm that found the DVGs. Datapane [39] and plotly [40] Python libraries where used to generate the HTML report and interactive plots, respectively. Examples of reports are available in the GitHub repositories associated with DVGfinder.

#### 2.7.2. Other Results

The information processed by DVGfinder is written as tables in csv files separated by the levels discussed in Section 2.7.1. (all, consensus, and filtered levels). Furthermore, the raw output of each search algorithm is kept in the directory “OldOutputs/virema” and “OldOutputs/ditector” in case users want to trace the events and retrieve other valuable information that these programs provide.

The alignment files and some stats about the alignments are also stored in “OldOutputs/alignments”.

### 2.8. Recommended Pipeline

The program can deal with complex RNA-seq data (host and virus reads), but a prior removal of host reads will significantly reduce the search time. In addition, the result of the program must be contextualized depending on the type of virus. In some RNA viruses, such as for SARS-CoV-2, translation of their structural proteins is performed from subgenomic fragments (sgRNA) [35,40,41,42,43]. These fragments will be identified as deletions by the program so if they are not of interest to the user, they should be filtered out of the output. In other cases, the exact identification of BP-RI coordinates will not be possible due to identical nucleotides in the proximity of the abnormal junction, and the programs identify multiple BP-RI coordinates. This situation has been reported by Mura et al. [44] and also in the synthetics and real datasets we have evaluated here. A strategy to be used in such case is presented in Section 2.9 below. An example of workflow can be found at Appendix A.

### 2.9. Evaluation Synthetic Datasets

To assess the performance of the tool we simulated two sets of 12 samples. Both sets were generated with the same DVG density (number of DVGs/genome length) from SARS-CoV-2 and turnip mosaic virus (TuMV) genomes, two (+)ssRNA viruses. The DVGs in the samples were defined randomly, each type of DVG (deletions, insertions and copy/snap-backs) being in equal proportions and supposing 60% of the total viral population in the samples. The rest of each sample was made up of the reference genome. More precisely, 216 and 72 random DVGs were introduced, respectively, and depth samples (*n*) of 10^5^, 5 × 10^5^, 10^6^, 2 × 10^6^, 3 × 10^6^, 4 × 10^6^, 5 × 10^6^, 6 × 10^6^, 7 × 10^6^, 8 × 10^6^, 9 × 10^6^, and 10^7^ of 100 nucleotide-long single-end reads were used. In total, we used 24 fastq simulated samples. A further description of the generation of these synthetic datasets is provided in Appendix A and the exact list of DVGs incorporated in each sample is given in Appendix A and in [45] and the fastq files in [46].

### 2.10. Assessment of the Program Performance with Synthetic Datasets

The independent results of ViReMa-a and DI-tector and the three output modes of DVGfinder (Metasearch, Consensus and Filtered) were compared on the above synthetic samples. The number of *TP*, *FP*, and *FN* could be quantified in each case.

In a first evaluation, we considered as *TP* only the DVGs for which cID_DI matched exactly with one of the DVGs introduced in the synthetic dataset or with its reverse complement (i.e, if “++_100_300” was in the list of reals, only the same DVG or “−−_300_100” was labeled as *TP*). The rest of the DVGs detected were counted as *FP*. Real events (DVGs introduced in the synthetic sample) not detected by the program were considered *FN*. A considerable amount of cID_DI qualified as *FP* but a junction position close to that of a real DVG was found with this strict evaluation [47]. Therefore, a second, less stringent evaluation was performed. For this evaluation, we proceeded as follows: (i) For each event identified by the program, we generated the sequence around the junction (30 nucleotides pre- and 30 nucleotides post-junction) considering its BP, RI, and DVG type. This size was chosen to avoid ambiguous matches with parts of the native genome. For shorter genomes it is recommended that the user adjust the length. (ii) Then, we searched whether this reconstructed sequence or its reverse complementary were present in any of the DVGs in the synthetic datasets. If so, then this DVG was requalified as *TP*; indeed, the BP and RI coordinates did not exactly match the original DVG. The results from this second evaluation are the ones presented and discussed in the Results section. The resulting tables, with the results separated by mode and the assigned category, are available at [46].

### 2.11. Statistical Evaluation

To assess the differences between the performance of the three DVGfinder run modes and the two searching programs separately, the Friedman rank sum test [48] was used. Whenever the null hypothesis was rejected, a pairwise comparison between the DVGfinder run modes and each one of the two published search programs was performed using a Wilcoxon signed rank test with a directional alternative, to check if the improvement was significant across all the library sizes simulated. All the *p* values were corrected by the Benjamini-Hochberg *FDR* method, imposing an overall significance level of 0.05. All the statistical tests were performed with R software version 4.0.2 in RStudio version 22 June 2020.

### 2.12. Assesment of the Program Performance with Real HTS Samples

To evaluate the performance of DVGfinder with a real HTS dataset, we used the data generated by Bosma et al. [28], consisting of four samples obtained from three mump virus preparations (one sample analyzed twice). The sequencing data are available at NCBI SRA under bioproject PRJNA525871 (files named as Virus_1_18C, Virus_2_19C, Virus_3_20C and Virus_2_HiSeq_rpt_11S, corresponding to runs SRR8719995 to SRR8719998, respectively). Twenty-eight copy-backs and one deletion were experimentally validated by RT-PCR by Bosma et al. [28].

We ran DVGfinder directly over the samples and compared the results with the validated DVGs. The only preprocess consisted of interleaving in a unique fastq each pair of paired-end reads, adding to the name of the reads a “/1” or “/2”, respectively. Since the sequencing method used did not discriminate among the RNA strands, we considered a validated DVG as detected if any of its direct or reverse complementary events was found.

## 3. Results

### 3.1. Evaluating DVGfinder Performance

We have evaluated the performance of the three DVGfinder run modes (Metasearch, Consensus, and Filtered), and the two searching programs (ViReMa-a and DI-tector), separately with the synthetic samples generated from SARS-CoV-2 and TuMV. As the synthetic samples had a known population of DVGs, *TP*, *FP*, and *FN* were calculated as detailed in Section 2.9, and sensitivity (*TPR*), precision (*PPV*), and *F*_1_ were calculated for each run mode. A visualization of the performance of ViReMa-a and DI-tector identifying DVGs can be found in Appendix A. The results are shown in Figure 5. Whether library size had an effect on the difference between the scores and run modes was also tested, first with a Friedman test and next with ad hoc analysis (Table 3).

The results have shown that there are significant differences between the run modes in all the scores and for both datasets (Table 3). In other words, in every multiple comparison exist, at least one run mode performs better than the others.

Pairwise comparisons between the DVGfinder run modes and the two search programs are presented and discussed in the following sections (Table 4).

#### 3.1.1. Sensitivity

Both search algorithms are very sensitive and only at smaller library sizes showed significant differences. In this scenario, DI-tector turns out to be more sensitive, yet DVGfinder Metasearch improves slightly the sensitivity (Figure 5A,B). The Metasearch DVGfinder’s mode achieved a significant improvement over ViReMa-a in the number of DVGs detected across all tested sizes of the SARS-CoV-2 dataset.

#### 3.1.2. Precision

The results obtained in terms of precision in both datasets are equivalent. The ViReMa-a, DVGfinder Filtered, and DVGfinder Consensus run modes achieved maximum precision on all samples, regardless of library size (Figure 5C,D). DI-tector shows a decay in precision with increasing sample size; consequently, DVGfinder Metasearch follows shows the same behavior although to a lesser extent due to the contribution of ViReMa-a.

As is summarized in Table 5, in both sets of samples, the three DVGfinder run modes reaches a significant better precision than DI-tector used alone.

#### 3.1.3. *F*_1_ Score

The *F*_1_ score, as a harmonic mean of sensitivity and precision, allows to evaluate the trade-off between these two values. In the smallest samples of each data set, DVGfinder Metasearch and DI-tector obtained the best scores, but with increasing sample size, the ViReMa-a and DVGfinder Filtered and Consensus modes reached the maximum value while the score of the first two declined. This behavior was observed in both data sets (Figure 5E,F).

The results in Table 6 show that the three modes of DVGfinder obtained a significant improvement compared to DI-tector.

### 3.2. DVGfinder Performance in Real Samples Containing Experimentally Validated DVGs

We selected a group of publicly available samples (SRR8719995–SRR8719998) generated from Vero cells infected with mump virus (−)ssRNA where a set of 28 5′ copy-backs were experimentally validated using DVG-specific RT-PCR primers [28]. DVGfinder was run on the three samples indicating to the program the antisense polarity of the reference virus. In total, DVGfinder founded 9453 DVGs in Sample 1, 11,758 in Sample 2, and 12,641 in Sample 3 (the raw output of the program can be found in [49].

The results obtained by the program were filtered to keep only the copy-backs (Appendix A) and then compared with the experimentally validated DVGs previously described (Table 7). With the Metasearch strategy, DVGfinder detected 24 of the 28 5′ copy-backs. This represents three more assignments than using ViReMa-a.

### 3.3. Final Report

DVGfinder generates an interactive report with the DVGs identified. An example of the resulting HTML report can be found in [50]. This example shows the results of the TuMV synthetic dataset generated as evaluation data, with a library size 100,000 reads. The reports generated from the real samples analyzed are available for exploration [51].

## 4. Discussion

There are two published DVG search algorithms that show a high sensitivity but also can incur a number of false positives. The strategy we have used in this work consisted of unifying, in a single application, the search strategies implemented in these two programs, also standardizing the terminology and trying to reduce the number of false positives by applying an ML classification algorithm. This is the first time that ML is implemented in a tool for the identification of DVGs in HTS data.

To evaluate the performance of the DVGfinder, we have generated two sets of synthetic samples based on the genomes of two (+)ssRNA viruses, SARS-CoV-2 and TuMV, which widely differ in their genome length, gene content, and expression mechanism. Each synthetic dataset was composed of a known and equal number of DVGs that included deletions, insertions, and copy- and snap-backs. The introduced DVGs were randomly selected, constituting 60% of the total sample reads; the rest of the synthetic sample was composed of the native viral genome. All reads were 100 nucleotides long and libraries of between 10^5^ and 10^7^ reads were generated. In total, 24 simulated fastq samples were used to assess the sensitivity, precision, and *F*_1_ score of the three DVGfinder run modes, as well as the two search programs separately.

These evaluations have shown that the DVGfinder Metasearch run mode significantly improved the sensitivity of ViReMa-a. This behavior was expected since the Metasearch run mode consists of the union of the DVGs identified by each algorithm. We observed that the sensitivity scores obtained for the 5 × 10^5^ reads TuMV samples are more similar to those obtained for the 1 × 10^6^ reads SARS-CoV-2 samples. This can be explained by the three-times shorter TuMV genome, which means that for the same library size, the average depth at each position is greater. Although further studies of the behavior of the search algorithms on different types of viral genomes would be needed, we believe that DVGfinder, as a metasearch engine, facilitates the benchmarking of the tools that integrate them. In addition, the DVGfinder architecture would easily allow to incorporate other search algorithms that might be published in the future.

Of the two previously published algorithms, ViReMa-a shows the best precision on the synthetic data presented. The Filtered and Consensus Modes of DVGfinder equal the scores of this tool and all three modes show a significant improvement in precision over DI-tector in the synthetic data.

Although the prediction model was trained with a dataset generated from SARS-CoV-2 genome, it works with other (+)ssRNA viral genomes, as TuMV. This could be because the features selected as predictors are not genome-dependent but instead depth-dependent; i.e., the metrics computed with the software and used as input for the ML classifier are related to the depth of the samples at the BP and RI coordinates and surrounding sequences.

Regarding the *F*_1_ score, that weights both sensitivity and precision values, we observed a significance improvement of all the DVGfinder modes over DI-tector in both synthetic datasets.

We have shown that DVGfinder deals well with real HTS data. We selected a set of publicly available samples of mumps virus for which a group of copy-backs were experimentally validated [28]. DVGfinder Metasearch strategy did a better job identifying the real copy-backs that ViReMa-a and also generates (without additional work by the user) interactive and easy-to-browse reports of the detected DVGs.

An important aspect to bear in mind is that the search programs integrated in DVGfinder identify candidate recombinant DVGs present in a sample, but do not provide any information about the potential role of these DVGs as defective interfering particles. To facilitate this study, a theoretical DVG length is calculated but we strongly recommend carefully considering the assumptions from which these reconstructions were done. On the one hand, DVGfinder takes the recombination event as the unique abnormal junction in the DVGs. On the other hand, it assumes that once the RdRp reattaches to the template, it continues polymerizing until running off at its extreme. These two simplifications could generate structures larger than the wild-type genome length (in some cases, even twice its size), the type of structures that should be considered as artifacts.

Lastly, the ViReMa-a output groups identified recombination events in the sense of the sequences before and after the recombination point. By contrast, the DI-tector output groups the possible recombinants by DVG type. We have homogenized the output and proposed a unique identifier with the form “sense_BP_RI”. This identifier provides into a single tag all the relevant structural information about any DVG and it is easy to reformat to a “start_end” identification in case we do not need to use the sense strand information.

## 5. Conclusions

Here, we present DVGfinder, a friendly and easy-to-use metasearch tool to identify DVGs in RNA-seq data. The program integrates the two most popular DVG search algorithms published in the last years, ViReMa-a and DI-tector, in a simple workflow. An ML model also was trained to sieve true positives from the total number of events identified by the program. The results of the program are written in various csv files with a unified terminology. In addition, an interactive HTML report, with summary tables and different plots, is generated to facilitate each possible DVG.

As long as the two search algorithms detect DVGs in the sample, the results are grouped in three run modes: the Metasearch mode shows the events detected; Consensus mode selects the intersected events; and the Filtered mode only shows the DVGs with a probability equal or higher than a user-defined threshold probability of true positives. In any case, the user can confirm in the output HTML which algorithm has identified the DVG.

We tested the performance of each run mode separately in two sets of synthetic samples based on the SARS-CoV-2 and TuMV genomes. The results show that the Metasearch run mode of DVGfinder has a significant improvement in sensitivity over ViReMa-a, while all the modes reached a better precision and *F*_1_ score than DI-tector. We could also verify that although the ML classifier was trained on a SARS-CoV-2 genome-based dataset, it also performs well with samples from other (+)ssRNA viruses as structurally different as TuMV. The results on the real HTS samples generated from the mump virus, a (−)ssRNA virus, shown that DVGfinder is a good option to use to identify DVG candidates.

## Figures and Tables

**Figure 1 viruses-14-01114-f001:**
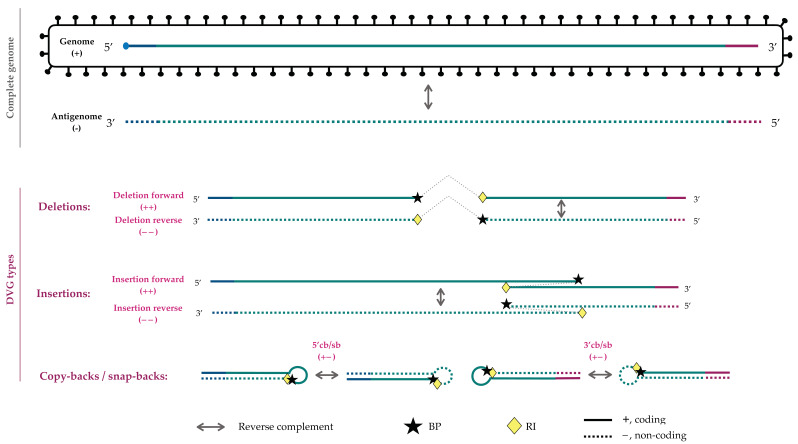
Schematic representation of the different types of DVGs that can be produced during a positive-sense RNA virus replication. In the case of negative-sense RNA virus, the terminology has to be reverted accordingly: coding (+) by non-coding (−), 5′cb/sb by 3′cb/sb, and “forward” by “reverse” and *vice versa*.

**Figure 2 viruses-14-01114-f002:**
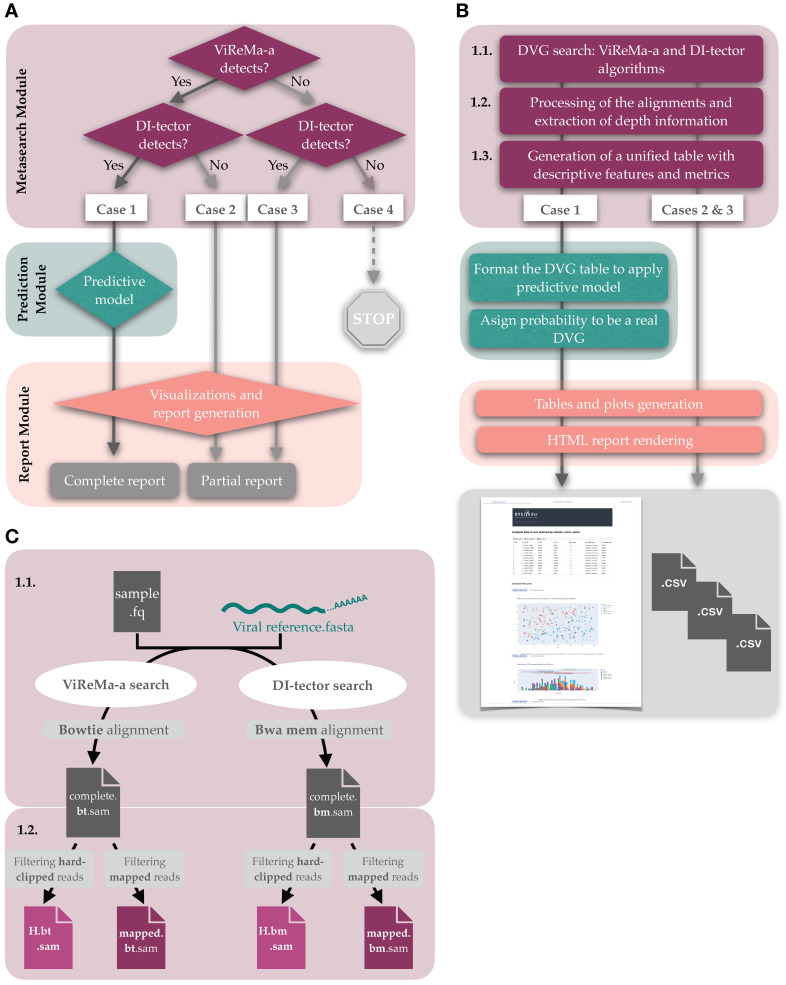
DVGfinder flow chart. (**A**) General workflow of the program with the four cases of usage accounted. (**B**) A more of detailed description. (**C**) Simplification of the two first steps of the Metasearch module, with emphasis on the processing of the different alignments obtained.

**Figure 3 viruses-14-01114-f003:**
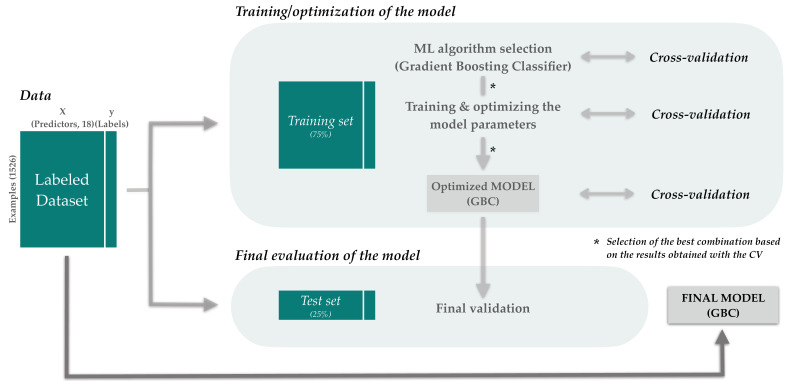
Model generation process. The labeled dataset, composed for a total of 1527 DVGs, between true positives (*TP* = 764) and false positives (*FP* = 762), was divided into two separate partitions of data (train and test set), respecting the proportion of the two categories. A GBC was chosen for training the model based on its best performance with respect to other algorithms such as Logistic Regression, Extra Trees Classifier, Supporting Vector Machine, and Random Forest. The optimization of the hyperparameters was achieved by the RandomizedSearchCV function of sklearn library using the training dataset. The optimized model was finally validated on the unseen test dataset considering that the model was generalizable. Finally, the model was trained with the totality of the data and integrated in the Predictive module of DVGfinder.

**Figure 4 viruses-14-01114-f004:**
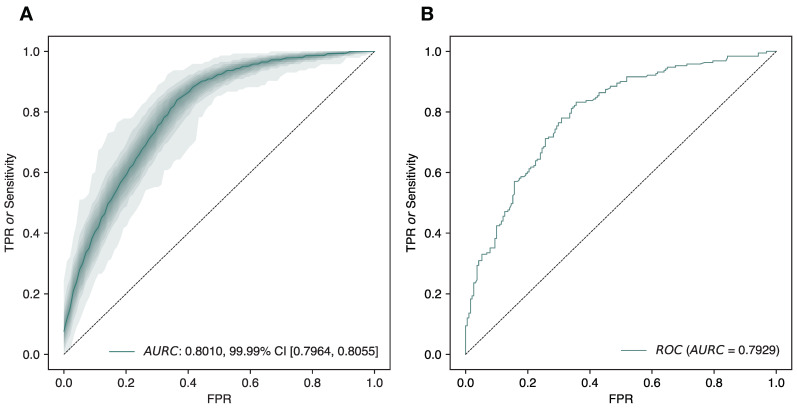
*ROC* curves of the optimized classifier (GBC algorithm). (**A**) Four-fold cross validation (100 repeats each fold) in the training set partition; the mean *ROC* is shown with a continuous line and the distribution of the data from 45–55 to 0–100 percentiles in ranges of 5% in the colored areas (from darker to lighter respectively). A mean *AURC* = 0.8010, with a 99.99% CI from 0.7964 and 0.8055, was achieved in the training set. (**B**) Final validation of the model in the test partition. Despite these data being completely new for the model, the *AURC* = 0.7929 was really close to the values of the training set, thus considering the model to be generalizable. The final model was trained with the whole dataset.

**Figure 5 viruses-14-01114-f005:**
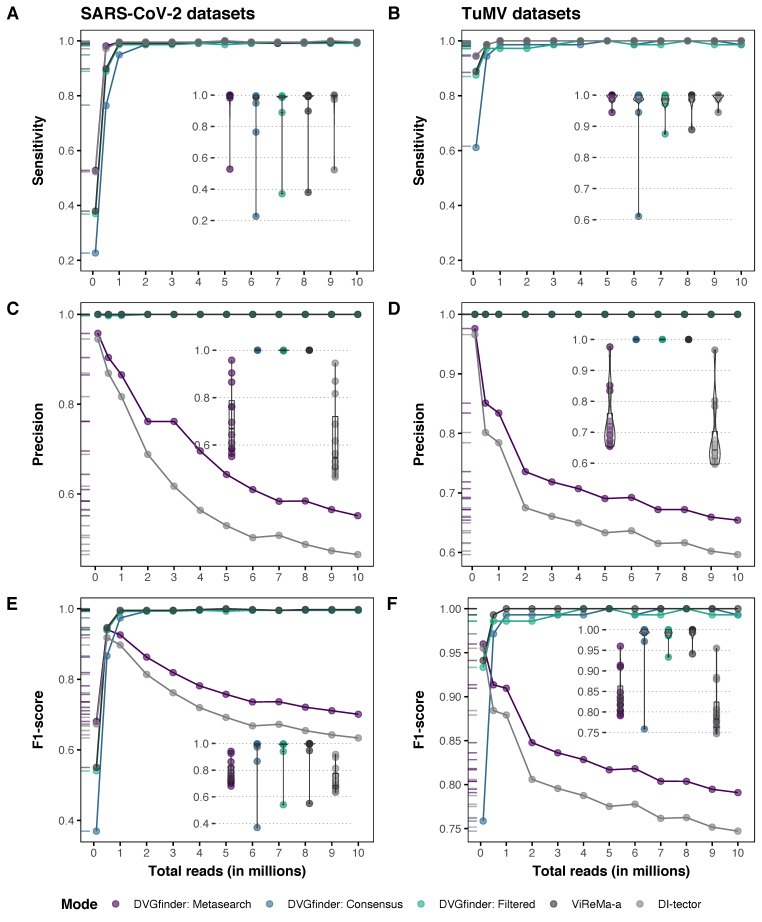
Performance of the three DVGfinder run modes (Metasearch, Consensus, and Filtered) and the two search algorithms (ViReMa-a and DI-tector) working separately on the synthetic samples generated from the SARS-CoV2 genome (**left column**) and TuMV genome (**right column**). Sensitivity (**A**,**B**), Precision (**C**,**D**), and *F*_1_ score (**E**,**F**) were calculated in the twelve synthetic samples generated from each genome. An identified DVG was considered as *TP* if the sequence around its junction matched with the sequence of any DVG introduced in the synthetic dataset, directly or as reverse complementary. The scores for each run mode are represented in three different types of plots: line graphs show them in relation to the library size while box and rug plots show the distribution of each mode. Within each genome set, each sample contains the same composition in number and type of DVGs, the size of the library being the unique parameter of variation. The library sizes assessed (number of reads) were 10^5^, 5 × 10^5^, 10^6^, 2 × 10^6^, 3 × 10^6^, 4 × 10^6^, 5 × 10^6^, 6 × 10^6^, 7 × 10^6^, 8 × 10^6^, 9 × 10^6^, and 10^7^.

**Table 1 viruses-14-01114-t001:** Published tools for the identification of DVGs. Abbreviatures: D, deletion; I, insertion; cb/sb, copy- and/or snap-back. The last column enumerates the studies that have used each program (search on PubMed database, last accessed on 23 February 2022).

Program	Language	DVG Type Identified	Reference	Number of Citations	Used by
ViReMa-a	Python	D, I, cb/sb	[14]	31	[16,17,18,19,20,21,22,23,24,25]
DI-tector	Python	D, I, cb/sb	[15]	9	[26,27,28,29,30]
VODKA	Perl	cb/sb	[31]	13	[32]
DVG-profiler	C++	D, I, cb/sb	[28]	4	-
DG-seq	R	D, I, cb/sb	[33]	5	-

**Table 2 viruses-14-01114-t002:** Coordinates used for the reconstruction of the theoretical length of the DVGs in a (+)ssRNA viral genome. The length is calculated as the sum of the nucleotides that conform the pre-BP and post-RI fragments.

DVG Type	Fragment	Sense	Coordinates Extracted from Reference
Deletion forward and insertion forward	pre-BP post-RI	+ +	[start, BP] [RI, end]
Deletion reverse and insertion reverse	pre-BP post-RI	− −	[BP, end] [start, RI]
5′ cb/sb	pre-BP post-RI	+ −	[start, BP] [start, RI]
3′ cb/sb	pre-BP post-RI	− +	[BP, end] [RI, end]

**Table 3 viruses-14-01114-t003:** Comparation of the five modes in the matched samples (twelve for each dataset) using the Friedmann test. In all the scores and datasets there were significant changes detected.

**SARS-CoV-2 Dataset**		
**Performance Index**	**Friedman Statistic**	** *p* **
Sensitivity (*TPR*)	40.6030	3.248 × 10^−8^
Precision (*PPV*)	46.7879	1.688 × 10^−9^
*F*_1_ score	30.6725	3.570 × 10^−6^
**TuMV Dataset**		
**Performance Index**	**Friedman Statistic**	** *p* **
Sensitivity (*TPR*)	30.7887	3.381 × 10^−6^
Precision (*PPV*)	39.6123	5.206 × 10^−8^
*F*_1_ score	32.9735	1.210 × 10^−6^

**Table 4 viruses-14-01114-t004:** Pairwise comparisons (Wilcoxon signed rank test for paired samples) for sensitivity in both sets of synthetic samples. The 1-tailed contrast performed is shown in the respective column. The *p*-value was corrected with Benjamini-Hochberg *FDR*.

**SARS-CoV-2 Dataset**			
**Performance Index**	**Contrast**	** *p* **	** *FDR* **
Sensitivity (*TPR*)	Metasearch-ViReMa-a	0.0089	0.0153
Sensitivity (*TPR*)	Metasearch-DI-tector	0.1855	0.2783
**TuMV Dataset**			
**Performance Index**	**Contrast**	** *p* **	** *FDR* **
Sensitivity (*TPR*)	Metasearch-ViReMa-a	0.5000	0.8571
Sensitivity (*TPR*)	Metasearch-DI-tector	1.0000	1.0000

**Table 5 viruses-14-01114-t005:** Pairwise comparisons (Wilcoxon signed rank test for paired samples) for precision in both sets of synthetic samples. The one-tailed contrast is defined in the contrast column. The *p*-value was corrected with Benjamini-Hochberg *FDR*.

**SARS-CoV-2 Dataset**			
**Performance Index**	**Contrast**	** *p* **	** *FDR* **
Precision (*PPV*)	Consensus-ViReMa-a	1.0000	1.0000
Precision (*PPV*)	Consensus-DI-tector	0.0013	0.0038
Precision (*PPV*)	Filtered-ViReMa-a	0.9632	1.0000
Precision (*PPV*)	Filtered-DI-tector	0.0013	0.0038
Precision (*PPV*)	Metasearch–DI-tector	0.0013	0.0038
**TuMV Dataset**			
**Performance Index**	**Contrast**	** *p* **	** *FDR* **
Precision (*PPV*)	Consensus-ViReMa-a	1.0000	1.0000
Precision (*PPV*)	Consensus-DI-tector	0.0013	0.0038
Precision (*PPV*)	Filtered-ViReMa-a	1.0000	1.0000
Precision (*PPV*)	Filtered-DI-tector	0.0013	0.0038
Precision (*PPV*)	Metasearch–DI-tector	0.0013	0.0038

**Table 6 viruses-14-01114-t006:** Pairwise comparisons (Wilcoxon signed rank test for paired samples) for *F*_1_ score in both sets of synthetic samples. The one-tailed contrast is shown in the contrast column. The *p*-value was corrected with Benjamini-Hochberg *FDR*.

**SARS-CoV-2 Dataset**			
**Performance Index**	**Contrast**	** *p* **	** *FDR* **
*F*_1_ score	Consensus-ViReMa-a	0.9981	1.0000
*F*_1_ score	Consensus-DI-tector	0.0067	0.0135
*F*_1_ score	Filtered-ViReMa-a	0.9980	1.0000
*F*_1_ score	Filtered-DI-tector	0.0027	0.0064
*F*_1_ score	Metasearch-DI-tector	0.0013	0.0038
**TuMV dataset**			
**Performance Index**	**Contrast**	** *p* **	** *FDR* **
*F*_1_ score	Consensus-ViReMa-a	0.9966	1.0000
*F*_1_ score	Consensus-DI-tector	0.0034	0.0068
*F*_1_ score	Filtered-ViReMa-a	0.9973	1.0000
*F*_1_ score	Filtered-DI-tector	0.0016	0.0039
*F*_1_ score	Metasearch-DI-tector	0.0013	0.0038

**Table 7 viruses-14-01114-t007:** 5′ copy-back DVG genomes in mumps virus samples validated using RT-PCR [28] and its in silico detection. In green are highlighted the DVGs that were detected only by DI-tector. The (*) indicate the DVG was found with the deep assessment strategy explained in Section 2.9, although the coordinates may not match exactly.

Sample	BP	RI	Predicted Size	ViReMa-a	DI-Tector	DVGfinder Metasearch
1	13,811–13,812	14,697–14,696	2262	Found	Found	Found
	13,580	14,964	2262	Found	Found *	Found
	13,865	14,964	1941	Found *	Found *	Found *
	14,575	14,934	1261	Not found	Not found	Not found
	14,770	15,004	997	Found	Found	Found
	14,874	15,104	792	Found	Found	Found
2	13,308	14,864	2598	Found	Found	Found
	13,566–13,568	14,678–14,676	2526	Found *	Found	Found
	13,617	14,633	2490	Not found	Not found	Not found
	13,316	15,144	2310	Not found	Found	Found
	13,339	15,131	2300	Found *	Found *	Found *
	13,347	15,239	2184	Not found	Found *	Found *
	13,480	15,064	1905	Found	Found	Found
	13,907–13,909	15,279–15,277	1584	Found	Found	Found
	14,456	14,885	1429	Not found	Found	Found
	14,223–14,224	15,166–15,165	1381	Found	Found	Found
	14,342	15,108	1320	Found	Found	Found
	14,596	14,896	1278	Not found	Not found	Not found
	14,761	14,917	1092	Found	Found	Found
	14,730–14,733	15,026–15,023	1014	Found	Found	Found
	14,868	15,032	870	Found	Found	Found
	14,856–14,863	15,043–15,036	870	Found *	Found	Found
	14,861–14,862	15,178–15,177	731	Found *	Found *	Found *
	14,947–14,950	15,145–15,142	678	Found	Found	Found
	14,864	15,198	708	Found	Found	Found
	14,934	15,272	564	Found	Found	Found
3	14,706	14,930	1134	Found	Found	Found
	12,661	15,231	2878	Not found	Not found	Not found
				21/28	24/28	24/28

## Data Availability

All scripts developed in this work are available at https://github.com/MJmaolu/DVGfinder, last accessed on 29 April 2022.

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
