# Peer review of "DVGfinder: A Metasearch Tool for Identifying Defective Viral Genomes in RNA-Seq Data"

_viruses, 2022, doi:10.3390/v14051114_

Round 1
Reviewer 1 Report
I have no further comments.
Author Response
Thank you so much for your review and suggestions. Very much appreciated.
Reviewer 2 Report
In this work the authors introduce a new tool, DVGfinder, whose aim is to find defective viral genome particles (DVGs) starting from high-throughput sequencing data. DVGfinder is based on ViReMa-a and DI-tector, two well-known tools used to detect DVGs, and it is designed both to refine their performance through a filtering module, and to allow an easy visualization of the results. The author's work also compares the performances of ViReMa-a, DI-tector, and of the different usage modes of DVGfinder, on both synthetic data produced by the authors and experimental data taken from the literature.
I think DVGfinder could be a nice addition to the bioinformatic toolbox for researchers interested in DVGs. In particular, beside slightly improving the precision and accuracy of ViReMa-a and DI-tector in specific situations, I think the final report module is particularly interesting, as it can be used to guide further research and to compare the results of the two DVG-detection tools on the same data. Moreover, the comparison among ViReMa-a and DI-tector discussed by the authors is of value and can be useful to many readers. Finally, I sincerely appreciate the efforts made by the authors to make open and easily accessible their code, their data, and everything else needed to reproduce their results.
However, I think that the author's work should be further improved before publication, and to do this I suggest to address the following issues:
1) In the abstract, the sentence “We evaluated […] coverage HTS.” in lines 21-23 could be misleading: for large datasets/high coverage DVGfinder does not outcompetes DI-tector in sensitivity, and ViReMa-a in precision. The effect for low coverage do exists, but it is quite small. So the authors should modify this sentence by toning down their claim.
2) In lines 30-31, the sentence “owed to the lack of proof-reading activities in their RNA-dependent RNA polymerases” is partially wrong: Coronaviruses are known to have a proof-reading mechanism (see e.g. the review by Sola et al., DOI: 10.1146/annurev-virology-100114-055218), but still the presence of defective copies of the viral genome has been documented and discussed in the literature. The authors should change this sequence accordingly.
3) Some recent papers should be cited by the authors: the work by Kim et al., DOI: 10.1016/j.cell.2020.04.011, where the authors use the aligner STAR to find defective viral genomes as non-canonical eukaryotic splicing events; when discussing the subgenomic RNAs (sgRNA) produced during SARS-CoV-2 replication, in lines 282-284, the authors should cite the work by Di Gioacchino et al., DOI: 10.1261/rna.078969.121 where a tool based on DI-tector is introduced to find sgRNAs, in addition to the citation to work by Parker et al., ref. 33 in the present version of the manuscript (where a tool is introduced to find sgRNAs for nanopore reads).
4) In lines 318-321 the authors discuss their strategy to deal with ambiguous BP/RI positions. Although interesting, the strategy used has a relevant shortcoming: 30 nucleotides are too many for reads of length 100 nucleotides, and may lead to biased performance assessments. I think 10 nucleotides for each side of the junction should be already enough (this would corresponds to a very small probability of introducing a false positive). So the authors should reduce this value from 30 to 10, or explain why they think 30 is a good number of nucleotides to consider. Moreover, in lines 432-433 they use another strategy to deal with the same ambiguity problem, i.e. they allow for a number of nucleotides of flexibility in the BP/RI determination. I do not understand why two different ways of dealing with the same problem are introduced, and the authors do not discuss this. So they should explain it better in the text, or choose one strategy and always use it.
Author Response
Thank you very much for your time and useful comments.
We appreciate your suggestions for adding additional references. All have been we incorporated into the revised version of the manuscript.
Regarding the strategies we used to evaluate the TPs, we wanted to show two of them. The first one, more complex computationally, was based on extract sequence identity. The second one, which allows to accommodate a number of sequence differences results in flexibility in the identification of the junction coordinates. Perhaps we were not clear enough in our explanation. Now we present the first strategy for both assessments. The results did not change in this case.
Regarding your comment about the size of the reconstructed sequence, we tried different lengths and selected 30 nucleotides per side because with shorter sequences some ambiguity appeared (mostly for the SARS-CoV-2 datasets). For example, using a 20-nucleotides long word (symmetrical around the junction), some of the detected events matched sequences from the native genome, not around the abnormal junction of the DVGs introduced in the synthetic dataset. This is, e.g. the case of the deletion “++_2044_3506” found by DI-tector in the 100K sample. The resulting sequence using 10 nucleotides “GGCTACTAACaatgccatgc” corresponds to positions 3495-3515 of the full genome. If we use a longer word, the match with the full genome disappears, revealing “++_2044_3506” as a FP.
Following your recommendation, we repeated the evaluation with the 10-nucleotides long. The precision doesn’t change significantly (see table below). The few changes are equivalent to the example discussed in the previous paragraph. In any case, the sensitivity did not suffer any variation, so we think the 30-nucleotides length is more reliable, at least for the SARS-CoV-2 based datasets and have decided to maintain it in the manuscript.
|
|
100K |
500K |
1M |
2M |
3M |
4M |
5M |
6M |
7M |
8M |
9M |
10M |
|
ViReMa-a |
1.0 |
1.0 |
1.0 |
1.0 |
1.0 |
1.0 |
1.0 |
1.0 |
1.0 |
1.0 |
1.0 |
1.0 |
|
DI-tector |
0.96 |
0.88 |
0.84 |
0.72 |
0.66 |
0.61 |
0.57 |
0.55 |
0.55 |
0.53 |
0.53 |
0.51 |
|
DVGfinder: Metasearch |
0.97 |
0.91 |
0.88 |
0.79 |
0.73 |
0.68 |
0.65 |
0.62 |
0.62 |
0.60 |
0.60 |
0.58 |
|
DVGfinder: Consensus |
1.0 |
1.0 |
1.0 |
1.0 |
1.0 |
1.0 |
1.0 |
1.0 |
1.0 |
1.0 |
1.0 |
1.0 |
|
DVGfinder: Filtered |
1.0 |
1.0 |
1.0 |
1.0 |
1.0 |
1.0 |
1.0 |
1.0 |
1.0 |
1.0 |
1.0 |
1.0 |
This manuscript is a resubmission of an earlier submission. The following is a list of the peer review reports and author responses from that submission.
Round 1
Reviewer 1 Report
In viruses-164417 Olmo-Uceda et al. propose a novel software (DVGfinder) that combines two most commonly used existing algorithms (ViReMa-a and DI-tector) with a new bioinformatic pipeline to detect defective viral genomes (DVG) from next-generation sequencing (NGS) data. Olmo-Uceda et al. study claims that DVGfinder increases the precision of ViReMa-a and DI-tector.
Developing of a large spectrum of bioinformatic algorithms is important to detect DVGs from NGS data obtained by various type of protocols. However, several major and minor points should be address before publishing this work.
Major points:
- The preparation of synthetic datasets should be explained in detail. The authors do explain this point in several parts of the manuscript (e.g. Sec. 2.4.1), but without giving enough details to allow to independently replicate their numerical experiments.
- Since Olmo-Uceda et al. study never deals with real data, the synthetic-data generation protocol is of extreme relevance to understand whether these new results are interesting for the case of real sequencing data. In particular, it seems that the authors' synthetic datasets were generated by putting together reference-genome synthetic reads and DVG synthetic reads, without introducing any sort of errors (e.g. technical errors). Analysis of real data by DVGfinder should be provided.
- According to Olmo-Uceda et al. study results, a tool such as DI-tector can give rise to false positives (notice that from Fig. 5 it seems that 60% of the DI-tector hits are false positives!): the first step of the tool consists in aligning all reads against the viral genome, and so perfect reference-genome synthetic reads will result in perfect (local) alignments, and as such, they will be neglected while searching for DVG reads. However, our autonomous verification of DI-tector algorithm on the independently generated synthetic dataset composed of perfect reference-genome synthetic reads and DVG synthetic reads resulted in 100% of DI-tector accuracy (as expected). Thus, the authors should better explain which are the origins of the errors that their algorithm is correcting, and how the errors are introduced into the synthetic dataset they generated.
- 5 illustrates the central results of the paper, but it is of very poor quality: (i) no legend is provided, making it impossible to understand (and to be sure of) what the plotted quantities refer to; (ii) it seems that the plots in the inset are box and violin, rather than box and rug plots; (iii) the x-axis is wrong, as the numbers are clearly not the library size (in millions). Therefore, the authors should improve this figure to make it understandable. Once this will be done, another round of review will be needed to assess the results presented here, given the importance of this figure.
Minor points:
- The authors seem to link BP/RI sites number with the use RPHT (reads per hundred thousand reads mapped to the reference genome) which hasn’t been previously documented as a longer viral genome not necessarily having more BP and RI sites.
- How does the resulting change if the RPHT is used for the classifier after multiplication by the length of the reference genome (to drop the dependency on its length)? The authors should answer these questions in their work.
- The four quantities used by the authors to distinguish between true and false positives depending on the exact BP and RI positions. However, for several viruses the exact BP and RI positions cannot be obtained from the mapped read due to identical nucleotides or even motifs present in different genome regions. In particular this happens for 5’-copy back DVG produced by the measles virus (Mura et.al J.Virol 2017) or for SARS-CoV-2 subgenomic RNA organized similarly to deletion form of DVGs (Di Gioacchino et al., RNA 2022). The authors should explain how does their algorithm deal with these situations.
- From Fig. 2A it seems that only when both ViReMa-a and DI-tector detect DVGs the predictive module is run to classify true and false positives. If this is indeed the case, the authors should explain why in the text.
- The authors should provide the viral genome IDs of each viral genome used in this work, to allow reproducibility of their results.
- The tutorial on the Github repository should be made closer to a real usage case, using one fixed virus to generate the synthetic dataset and as a reference genome (now it seems that TuMV has been used to generate the synthetic dataset, and then SARS-CoV-2 is used as the reference viral genome).
- The first sentence in the abstract (“The generation of different types of defective viral genomes (DVG) is an unavoidable consequence of the error-prone replication of RNA viruses.”) should be corrected since no validated mechanism of DVG production has been justified as later discussed in the same manuscript.
- Line 33 “non-infectious defective viruses” should be changed to “defective viral particles”.
Author Response
Please see the attachement.

Reviewer 2 Report
This manuscript aimed to develop a pipeline using the two widely used algorithms, ViReMa and DI-tector, to better facilitate DVG identification from deep sequencing dataset. I think this type of work certainly is very useful and necessary to improve the DVG field. However, the big concern I have for this manuscript is that the authors only used simulated dataset to test the pipeline (for both training and testing). And the simulated dataset only contains DVGs and viral full-length genomes, no host and no other noise reads, which are not the case in real RNA-seq dataset. Although the data suggest DVGfinder is better than ViReMa and DI-tector for identifying true DVGs in terms of sensitivity and precision, whether it is better for real RNA-seq data from in vitro infection and even clinical samples is questionable. In addition, as the simulated dataset focused on deletion DVGs, no data indicate how this pipeline performs in terms of identifying copy-back DVGs.
Major point:
The simulated dataset is over simplified to test this pipeline. I suggest that the authors add host sequences and other non-host, non-targeted viral reads in the simulated dataset and test DVGfinder. Alternatively, have the authors tested DVGfinder on an RNA-seq data from a real infection or clinical samples? If so, how do the results look like compared to ViReMa?
Minor suggestion:
- I think the authors need to be more specific about the deep sequencing data that DVGfinder can apply to. For example, what kind of deep sequencing data, short-reads or long-reads, bulk RNA-seq or scRNA-seq? If short read, what is the range for the length of reads? The simulated dataset is 100nt long and single-ended and are those conditions the only conditions the authors recommend users to use for DVGfinder?
- As the simulated dataset did not contain cellular or other host reads, do the authors recommend to filter out the host reads before applying DVGfinder? Have the authors tested this?
- In the final results, will the actual read sequence and read ID be reported? I think it is useful and important to report these information in DVGfinder.
Author Response
Please see the attachement.

Round 2
Reviewer 1 Report
We are not convinced by answer A.1.3. Actually, after careful reading of that answer, and additional tests we performed, we believe that the authors are using DI-tector in an incorrect way, and that the vast majority of the false positives they obtain, and that their tool has been designed to correct, is due to their misuse of the tool. Since the authors uploaded their synthetic reads, we can now provide an explicit example, a read taken from the file sars216_N500K_l100.fq . The ID of the read is @Deletion_reverse_27968_1294_1599_1956_0:0:0_0:0:0_3/1 . With this read as a working example, we can discuss two fundamental mistakes done by the authors.
- The actual junction of the read above, as taken from the file sars216_N500K_l100_composition. csv , joins positions 1294 and 27968 of the reference genome. DI-tector, however, returns a junction in positions 1295 and 27969. The first problem of the work, which the authors state but apparently did not investigate deeply enough, is that this answer corresponds to a false negative in their performance estimation. Now, in both cases (junction in 1294/27968 or in 1295/27969), the resulting read would be identical , so there is absolutely no way of discriminating a read with a junction in positions 1294/27968 and another with a junction in position 1295/27969. For this reason, from a biological point of view, it makes no sense to consider the junction found in 1295/27969 as a false positive (we stress, once more, the fact that this junction exists in the fq file!). This happens actually for several reads, and the length of ambiguous nucleotides (i.e. nucleotides that can be, equivalently, assigned to the 5' or 3' side of the junction) can be quite high (up to 6 for the deletions obtained by DI-tector run with standard parameters on the file sars216_N500K_l100.fq ). Therefore, one of the following two solutions should be used to address this problem: 1) allow for certain flexibility when deciding if a junction found is a false positive or a true positive; 2) post-process each junction (both the ground truth and those obtained from the DVG-detection tools) so that a fixed choice is made whenever ambiguous nucleotides are present, such as fixing the BP position is as close as possible to the 5' end of the genome (see, for instance, the approach used here https://github.com/adigioacchino/sgDI-tector for deletions). Although this second option is the best, the first is much faster to implement, so we will report below the results of tests made using the first option.
- The second issue is that in the authors' notations, the read above has 27968 as BP and 1294 as RI. DI-tector, on the opposite, reports position 1295 as BP and 27969 as RI. This (the swap of BP and RI) is just a matter of notation, and cannot be considered as a false positive due to DI-tector's algorithm! To confirm this, indeed, we checked that, for deletions of the sars216_N500K_l100.fq file the precision of DI-tector increases sensibly when BP and RI of reverse deletions are swapped. Again, to correct this mistake, the best option would be to use a notation for the BP and RI position which is consistent across the ground truth file and the tools used, but a quick and simple workaround, that we will use for tests below, is to check each time for both the BP/RI found in the ground truth file and the swapped case.
We did a quick test on the file sars216_N500K_l100.fq , by correcting the two problems above with the fastest methods (so allowing for a flexibility of 6 nucleotides in the determination of BP and RI, as well as considering as a true positive, for each BP/RI in the ground truth file, also the pair where they are swapped). The results are 98% of sensibility and 97% of precision for DI-tector, while the authors report something around 55% of both for the same file.
To conclude we think these two relevant mistakes made by the authors are too relevant to allow for the publication of their tool. Indeed, the performances of DI-tector are wrongly estimated (and a similar problem could be present for ViReMa). Therefore, for us, the usefulness of the tool proposed by the authors is not clear anymore.
Reviewer 2 Report
I am satisfied with the authors' responses and revisons.